# Variation in COVID-19 booster uptake in England: An ecological study

**Greg Dropkin** (ID) *

Independent Researcher, Liverpool, England

* gregd@gn.apc.org

**Data Availability Statement:** All relevant data are within the manuscript and its Supporting information files, and the National Statistics Postcode Lookup is linked from the manuscript.

**Funding:** The author received no specific funding for this work.

## Abstract

### Introduction

Variable and low uptake of the COVID-19 booster is a recognised problem, associated with individual characteristics including age, gender, ethnicity, and deprivation. Are there other relevant predictors at area level?

### Methods

Anonymous grouped data was downloaded from the UK Government Coronavirus Dashboard for Middle Super Output Areas (MSOA) in England, along with demographic, employment, and health data from public sources. Mixed models with a random intercept for Upper Tier Local Authority were analysed as quasibinomial Generalized Additive Models. The estimated random effects were then fitted with Bayesian linear mixed models using flu vaccination uptake, change in public health budgets, population proportion of vaccination sites at pharmacies, GP-led, vaccination centres, and hospital hubs, and Region.

### Results

Models for the MSOA-level COVID-19 first and second vaccinations and the Third Injection (including the booster), fit well. Index of Multiple Deprivation, proportion Aged 15-24 and 25-44, and ethnicity groupings Other White, Indian-Pakistani-Bangladeshi, and African-Caribbean-Other Black-Other, are highly significant predictors of lower uptake. The estimated random effects vary widely amongst local authorities, with positive impact of flu vaccine uptake and change in public health budgets, and regional impacts which are positive for London and South East (first and second doses only), and negative for North West and North East. The impact of vaccination sites did not reach 90% credibility, in general.

### Conclusion

COVID-19 vaccination rates at each stage are very well modelled if local authority random effects are included along with non-linear terms for demographic, employment and health data. Deprivation, younger age, and Other White, South Asian, and African-Caribbean-Other ethnicities are associated with lower uptake. The estimated local effects show strong regional variation and are positively associated with flu vaccination and increasing public

**Competing interests:** The authors have declared
that no competing interests exist.

health budgets. One simple way to improve COVID-19 vaccine uptake in England would be
to increase local public health allocations.

## Introduction

The booster programme has been central to the UK government's strategy for containing
COVID-19 in England during the autumn and winter of 2021–22. The booster was introduced
on 16 September 2021, and rolled out to people in eligible groups who had their second pri-
mary dose at least six months previously [1]. However, uptake of the booster remains well
below the levels achieved for the first and second vaccine doses [2]. Vaccination uptake is
highly dependent on age, gender, ethnicity, and deprivation, as is widely recognised both in
the UK [3–7] and internationally [8, 9].

As demography varies across England, vaccination rates will also vary. For example, as the
Omicron BA.1 variant swept the country on 4 January, the cumulative Third Injection
(Booster or 3rd primary for persons with compromised immunity) uptake in Newham was
27.2%, in contrast with 41.1% in Liverpool and 67.4% in Gloucestershire, according to data
from the UK Coronavirus Dashboard [10]. These are different populations, but does demogra-
phy explain the disparity in full? If there are additional sources of variation, are they character-
istics of people or of where they live?

I explore these questions by modelling cumulative COVID-19 vaccination in 6789 Middle
Super Output Areas (MSOA) of England, a UK Census geographic unit comprising around
8,000 people on average and covering the whole of England. Each MSOA belongs to a single
Upper Tier Local Authority (UTLA), an adminstrative and geographic unit. There are 149
UTLA, containing an average of 46 MSOA each, ranging from 5 (Rutland) to 182 (Kent).
Apart from Rutland, each UTLA contains at least 12 MSOA. In the first stage of analysis, the
observed vaccinations at MSOA level are fitted with a non-linear mixed model using demo-
graphic, employment and health covariates at MSOA level and a random intercept for the
UTLA. The model does not estimate individual take-up of vaccinations on the basis of covari-
ates known for individuals, but accounts for the take-up at MSOA level using MSOA covari-
ates including the UTLA to which the MSOA belongs. This analysis is structured as a (non-
Bayesian) Generalized Additive Model, whose output includes curves showing the impact of
each covariate in the presence of all others, and an estimated random effect for each UTLA.

These estimated random effects are then taken as observations in a Bayesian linear mixed
model, using selected predictors available at UTLA level and a random intercept for the 9
Regions which partition the 149 UTLA. The fixed effects in this second stage model are flu vac-
cination rate, the population proportions of various types of vaccination sites, and annual
change in local authority public health budgets. Regional disparity, public experience of flu
vaccination, access to COVID-19 vaccines, and changes in public health department capacity,
are all plausible influences on how local authorities may differ even while controlling for other
factors present in the first stage model.

An overview of the modelling is set out in the Statistical methods below, with full codes and
simulations in the Supporting information. The first stage models fit the data well, and show
variation in Upper Tier Local Authority random effects. The second stage models identify pos-
itive and negative predictors for the estimated random effects.

## Materials and methods

### Data sources

For each MSOA, the cumulative number of persons vaccinated, separately for each of the first two primary doses and the Third Injection, is available daily from the UK Coronavirus Dashboard along with the Vaccine Register Population. Data for the Third Injection comprises people given the Booster plus people over 12 years old with severely weakened immune system given a third (primary) dose. The Dashboard also provides MSOA incidence ("7 day rolling rate") of COVID-19 cases, weekly through to 5 days before download. Dashboard data was downloaded on 4 January 2022.

For each MSOA, the Index of Multiple Deprivation (IMD), and separate indices for the Health and Education deprivation domains, were taken from the English indices of deprivation 2019 [11]. Each MSOA comprises a number of Lower Super Output Areas (LSOA), whose population and area (mid-2019 estimates) [12] were used to obtain population density at MSOA level. Population by MSOA was taken from the mid-2019 estimates [13] which include data by gender and each year of age. Ethnicity at MSOA level from the 2011 Census was downloaded from the Office for National Statistics (ONS) Nomis dataset DC2101EW—Ethnic group by sex by age [14]. Employment by industrial sector is available at MSOA level from the 2011 Census and was downloaded from the Nomis dataset DC6110EW [15]. Communal establishments data was downloaded from the Nomis dataset QS421EW [16], and Multi-occupation housing from the Nomis dataset DC1109EW [17].

Mean distance to GP, Emergency Department, and Pharmacy is available at LSOA level from the AHAH dataset [18]. GP registration mapped to LSOA is available from NHS Digital [19]. MSOA level values were obtained as population-weighted means from the LSOA values within the MSOA. A lookup from MSOA to Local Authority to Upper Tier Local Authority to Region was available from Public Health England as part of weekly surveillance. The 28 October 2020 report is online, with lookup in columns 1–8 (select all, show columns) [20].

Seasonal Flu Vaccine Uptake data for 2020/21 at UTLA level is released by Public Health England [21]. Data was extracted for GP-registered patients aged under 65. The uptake for "Leicester and Rutland" was then assigned to each area separately. COVID-19 vaccination sites as of 17 November 2021 are listed by NHS England and NHS Improvement [22], separately for pharmacies, GP-led, vaccination centres, and hospital hubs. Their postcodes were mapped to Lower Tier Local Authority codes with the ONS National Statistics Postcode Lookup [23], and then to UTLA codes via the lookup above. Public Health ring-fenced grant allocations for 2020/21 and 2021/22 are available at UTLA level from the Department of Health [24, 25].

A shapefile with the May 2021 UTLA boundaries was downloaded from the ONS Open Geography Portal [26]. ONS files are free to use without a specific application for a licence, subject to the conditions of the Open Government Licence and the Framework [27]. Source: Office for National Statistics licensed under the Open Government Licence v.3.0. Contains OS data © Crown copyright and database right [2022].

### Statistical method

The first stage of modelling seeks to account for the observed vaccination rates at MSOA level using demographic, employment and health data along with random effects for the 149 UTLA.

Cumulative numbers of persons vaccinated were combined with the Vaccine Register Population to form a matrix **vmat** with the numbers vaccinated and unvaccinated in each of 6789 MSOA, as of 4 January 2022. For each vaccination, the corresponding matrix (**vmat1** etc) is

the response variable. Four models were considered, differing in the choice of explanatory covariates, and fitted separately for each vaccination (first and second primary doses, and the Third Injection).

These first stage models were structured as Generalized Additive Models (GAM), using the R package "mgcv" [28–30]. The factor **utla** with 149 levels denotes the UTLA for each MSOA, and is treated as a random effect in all four models. All other covariates are treated as fixed effects.

In one sense, these particular models have a simple form, because the random effect of **utla** is a random intercept, without interaction with the other covariates. A model of this type is the simplest of the mixed models outlined within a linear context in a review of Multilevel Analysis in Public Health Research [31] (Equation 4 in this reference would reduce to $Y_{ij} = \gamma_{00} + \gamma_{10} I_{ij} + U_{0j} + \epsilon_{ij}$). Random intercept models are included in a more recent review of ecological modelling [32], in a linear or generalized linear context. However, it is overly restrictive to assume models of these types. Non-linear mixed models are included in Mixed Effects Models in S and S-PLUS [33]. GAM is an even more flexible approach, in which the transformed mean response is approximated by unknown combinations of spline functions of the covariates.

Briefly, GAM models without interaction terms have the general form:

$$g(\mu_r) = X_r * \theta + f_1(x_{1r}) + \ldots + f_j(x_{jr}) \tag{1}$$

where $\mu_r = E(y_r)$ is the expectation of the $r^{th}$ response $y_r$ which is assumed to follow a specified distribution, $g$ is the link function, $X_r$ is the $r^{th}$ row of the model matrix for the parametric terms with parameter vector $\theta$, and the $f_j$ are smooth functions of the covariates $x_j$ whose value at the $r^{th}$ observation is $x_{jr}$.

In our case, $y_r$ is the vaccine uptake **vmat**[r,1]/(**vmat**[r,1]+**vmat**[r,2]) and $g$ is the standard link function for binomial data, $g(\mu) = log(\mu/(1 - \mu))$. The random intercepts appear as $f_1(x_{1r})$ where $x_1$ is the factor **utla** and $f_1$ maps each level of the factor to an unknown coefficient $c_i$ whose value is estimated during the fitting. The parametric term $X_r * \theta$ is omitted and all levels of **utla** enter the model on an equal footing.

The smooth functions $f_2 \ldots$ are splines, evaluated at the covariates for the fixed effects which were scaled or transformed before applying the spline functions. Covariates for IMD, cumulative incidence, and population density were scaled. Most other covariates were transformed to improve performance. For a covariate **x**, the monotonic transform **Lx** is defined as log(1+**x**/mean(**x**)). For example, the population proportion for a particular ethnicity or industrial sector may be very low, but its transformation has better distribution and will be more easily smoothed.

The vaccination data has greater variability than expected with a binomial distribution, so the models use the "quasibinomial" family, whose dispersion is estimated during fitting. Models use cubic regression splines with basis dimension 7.

Model $\mathcal{A}$ uses a random intercept for **utla**, and smoothers for cumulative incidence (scaled), IMD (scaled), population density (scaled), five age bands (as transformed proportions), male (transformed proportion), and six ethnicities (transformed proportions). The age bands are 0–4, 5–14, 15–24, 25–44, and 45–64. The ethnicities, using the designations from the 2011 Census, are White Irish and White Gypsy (combined group); Other White; Indian, Pakistani, and Bangladeshi (combined group); Chinese and Other Asian (combined group); African, Caribbean, Other Black, and Other Ethnicity (combined group); Arab.

Model $\mathcal{B}$ enlarges $\mathcal{A}$ with additional terms for the transformed population proportions of 15 industrial sectors: Agriculture energy and water, Manufacturing, Construction, Wholesale

and retail, Transport and storage, Accommodation and food service, Information and communication, Financial and insurance, Real estate, Professional scientific and technical, Administrative and support service, Public adminstration and defence, Education, Human health and social work, Other.

Model $\mathcal{C}$ enlarges $\mathcal{A}$ with additional demographic and local health covariates for transformed population proportion of Communal establishments, of Multi-occupation housing, population weighted mean distance to GP, to A&E, to Pharmacy, proportion registered with GP, Health deprivation, and Education deprivation.

Model $\mathcal{D}$ enlarges $\mathcal{A}$ with most of the additional terms from $\mathcal{B}$ and $\mathcal{C}$, but omits six terms whose contributions showed negligible or low significance: population density, Manufacturing, Accommodation and food, Information and communication, Real estate, mean distance to GP. The included fixed effects cover cumulative incidence, IMD, age (5 bands), gender, ethnicity (6 groups), employment (11 sectors), communal establishments, multi-housing, distance to A & E, distance to pharmacy, GP registration, health deprivation, educational deprivation.

The methods used here for fitting a GAM and optimising the choice of smoothing parameters are outlined in S1 Appendix. Models were fitted separately for each of the first two doses and the Third Injection. Model fit was evaluated by residual plots, a test of convergence of the smoothness parameter optimisation, and tests of whether the basis dimensions are sufficient. Outliers were detected by Cooks distance $> 0.02$, a somewhat arbitrary criterion. The actual and predicted proportions vaccinated in each MSOA were plotted, along with the smoother curves for particular covariates. Models were compared by ML value (see S1 Appendix), with lower value indicating better fit. For each smoothed covariate, the fitted model shows a p-value for an approximate significance test against the hypothesis that the corresponding smoother is in fact zero [29, 30].

For each fitted model, the first 149 coefficients $c_i$ are the estimated contributions of the **utla** factor levels. These coefficients also enable comparison of the fitted value for a particular MSOA, with the hypothetical fitted value if the same population were located in a different UTLA, replacing $c_i$ with $c_j$. If $g$ denotes the link $g(\mu) = log(\mu/(1 - \mu))$ and $h$ is the inverse link $h(t) = 1/(1 + exp(-t))$, the fitted value would be altered from $fit1$ to $h(g(fit1) - c_i + c_j)$.

As outlined in S1 Appendix, each $c_i$ has an approximate standard error $se_i$. "Caterpillar plots" were drawn using the $c_i$ to show the range of estimated random effects, with approximate 95% credible intervals obtained from the $se_i$. A UTLA map of the variation across England was produced in QGIS [34], using the ONS boundary shapefile [26] with shadings determined by the $c_i$ values.

In a second stage of modelling, the $c_i$ were taken as observed, to be predicted using covariates available for Upper Tier Local Authorities. Region was treated as a random effect, with fixed effects for the uptake of Flu vaccination, the change in Public Health ring-fenced grant allocations from 2020–21 to 2021–22, and the number of each type of COVID-19 vaccination site, per population within the local authority.

For this stage, 4000 posterior draws from a linear mixed model were obtained with the Bayesian programme rstanarm [35], and graphical checks were carried out with shinystan [36] and bayesplot [37]. Output includes parameter estimates and 90% credible intervals. Plots of predicted and observed $c_i$ were generated for each dose and the Third Injection. Using the associated package loo [38], a pointwise value of Pareto k $> 0.7$ was taken to indicate an outlier [39]. A Bayesian version of $R^2$ describes overall model fit [40].

In fact the $c_i$ are not observed, but are amongst the coefficients of the fitted first stage model. The coefficients have an associated covariance $V_c$ (S1 Appendix). To estimate the impact of this uncertainty, simulated $\mathbf{c}^*$ were drawn as multivariate normal with mean $\mathbf{c}$ (the vector with components $c_i$) and variance $V_c[1:149, 1:149]$. Second stage modelling was

**Table 1. ML values and proportion of deviance explained.**

| Model | First dose | | Second Dose | | Third Injection | |
|---|---|---|---|---|---|---|
| | ML | DevExp | ML | DevExp | ML | DevExp |
| $\mathcal{A}$ | -15317.25 | 0.953 | -15905.70 | 0.958 | -15605.74 | 0.965 |
| $\mathcal{B}$ | -15695.38 | 0.959 | -16208.56 | 0.962 | -16012.88 | 0.969 |
| $\mathcal{C}$ | -15985.15 | 0.962 | -16503.18 | 0.965 | -15981.23 | 0.969 |
| $\mathcal{D}$ | -16198.07 | 0.965 | -16691.77 | 0.968 | -16222.98 | 0.972 |

For each model and vaccination, the table gives the ML value of the fitted model and the proportion of deviance explained (see S1 Appendix). Lower values of ML correspond to models with a better fit, according to the criterion used here to fit the models. Higher values of DevExp correspond to models which come closer to the best possible fit using a different criterion (the deviance).

repeated using the simulated $\mathbf{c}^*$ to give fresh sampling output (4000 rows), and this process was repeated 100 times to produce a combined sampling matrix with 404,000 rows (including the original output from second stage modelling of the $c_i$). The mean, 0.05 and 0.95 quantiles of its columns were taken as corrected estimates and credible intervals of the second stage parameters, taking into account the uncertainty in the $c_i$.

A guide, data, code files, simulations, summary of model $\mathcal{D}$ and the appendix are included as Supporting information.

## Results

Outputs for the 4 first stage models and 3 vaccinations are shown in Table 1. For each vaccination, $\mathcal{D}$ has the lowest ML value and the highest Proportion of deviance explained (see S1 Appendix). The models pass tests of model fit, and spline basis dimensions were sufficient (see Statistical methods).

For the Third Injection with model $\mathcal{D}$, all predictors have highly significant impacts (p<0.001) except for distance to pharmacy (p<0.005) (see S7 File). Smoothers for each of the following covariates have p<2e-16: **utla**, the age bands 5–14, 15–24, and 25–44, ethnicities Other White, African Caribbean Other Black and Other (combined), South Asian (combined), IMD, Male, GP Registration, Health and Social work. $\mathcal{D}$ explains 97.2% of the deviance, whilst a model using only the random effects term explains 51.5%, and a model using only the fixed effects explains 95.7% (percentages do not sum to 97.2% as in the absence of some terms, others compensate). Both of the reduced models have considerably worse fit than $\mathcal{D}$ (higher ML values), and are far inferior by anova tests. In particular, the fixed effects model has ML = -15247.41, which is 975.57 above the value for $\mathcal{D}$, and an anova test against $\mathcal{D}$ has p<2.2e-16, whilst the random effects term in $\mathcal{D}$ has p<2e-16. This is strong evidence that **utla** has a significant impact on vaccination uptake, even when controlling for all the fixed effects.

Figures were produced for the Third Injection with model $\mathcal{D}$. Fig 1 shows the observed and fitted values of uptake, the outliers with Cooks distance > 0.02 and the 37 MSOA within Newham. Cooks distance is a metric combining the influence of individual points on the fitted model, and the magnitude of their deviance residual. There are six outliers as judged by Cooks distance > 0.02, and of these only Forest Heath 002 (Lakenheath) has a high residual, the others being highly influential points. Newham 013 (Olympic Park & Mill Meads) is an outlier, with uptake higher than expected. Newham 035 (Beckton Park), with the lowest uptake (0.223) in this local authority, nearly matches its predicted value (0.229). Overall, the model fits the data very well.

Fig 2 shows selected smoothers for model $\mathcal{D}$. The smoothers in panels A—G have much larger impact on fitted values than subsequent covariates, and those in panels B—G have a

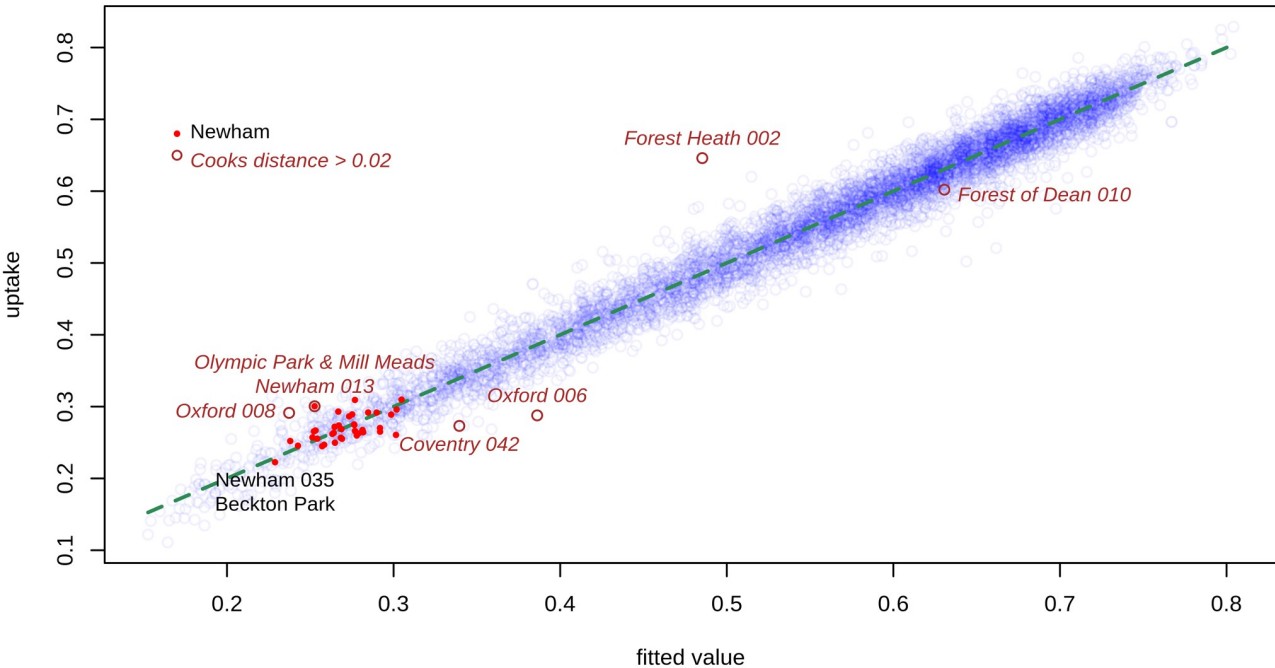

**Fig 1. Fitted value versus uptake, model $\mathcal{D}$ for the Third Injection.** Observed values of uptake are shown as semi-transparent blue points, and the green dashed line shows the fitted values. Outliers are shown in brown, and the Middle Super Output Areas within Newham in red. Vaccination data as of 4 January 2022. $\mathcal{D}$ is a Generalized Additive quasibinomial mixed model, with a random effect for Upper Tier Local Authority, and fixed effects as described in Statistical methods.

negative impact: higher values of scaled IMD (Fig 2B) give lower fitted values, and likewise for the age bands and ethnicities shown. An increasing male proportion (Fig 2H) also has a negative impact, but smaller than that of IMD, age, or ethnicity.

Fig 3 shows the "Caterpillar plot" for the estimated random effects $c_i$. Many estimated effects have 95% credible intervals which do not overlap the mean value. On the left, 51 estimated random effects are negative, whilst on the right 56 are positive. Only 42 UTLA have estimated effects whose 95% credible interval includes the mean value. The estimated random effects obtained from fitting model $\mathcal{D}$ to the first and second doses are highly correlated ($\rho = 0.99$), whilst the estimates when fitting to the second dose and Third Injection are well correlated ($\rho = 0.85$).

Fig 4 maps the geographic distribution of estimated random effects using model $\mathcal{D}$ for the Third Injection. There is clear regional variation, with negative impact for Upper Tier Local Authorities in the North West and parts of the North East, and positive impact in parts of London, the South East and South West.

Table 2 shows the results of second stage modelling of the scaled estimated random effects from each of four models for the Third Injection. This second stage modelling is then repeated with the scaled simulated random effects from each model for the Third Injection.

Bayes $R^2$ improves when second stage modelling is applied to the estimated random effects obtained from better first stage models ($\mathcal{D}$ is preferred to other models by ML value and leads to higher Bayes $R^2$). All four models lead to second stage models showing Flu vaccination and the rise in Public Health budgets as positive predictors (90% credible interval strictly positive) of estimated or simulated random effects. The population proportion of Vaccination Centres is a positive predictor only with model $\mathcal{C}$. North West Region is a negative predictor with all

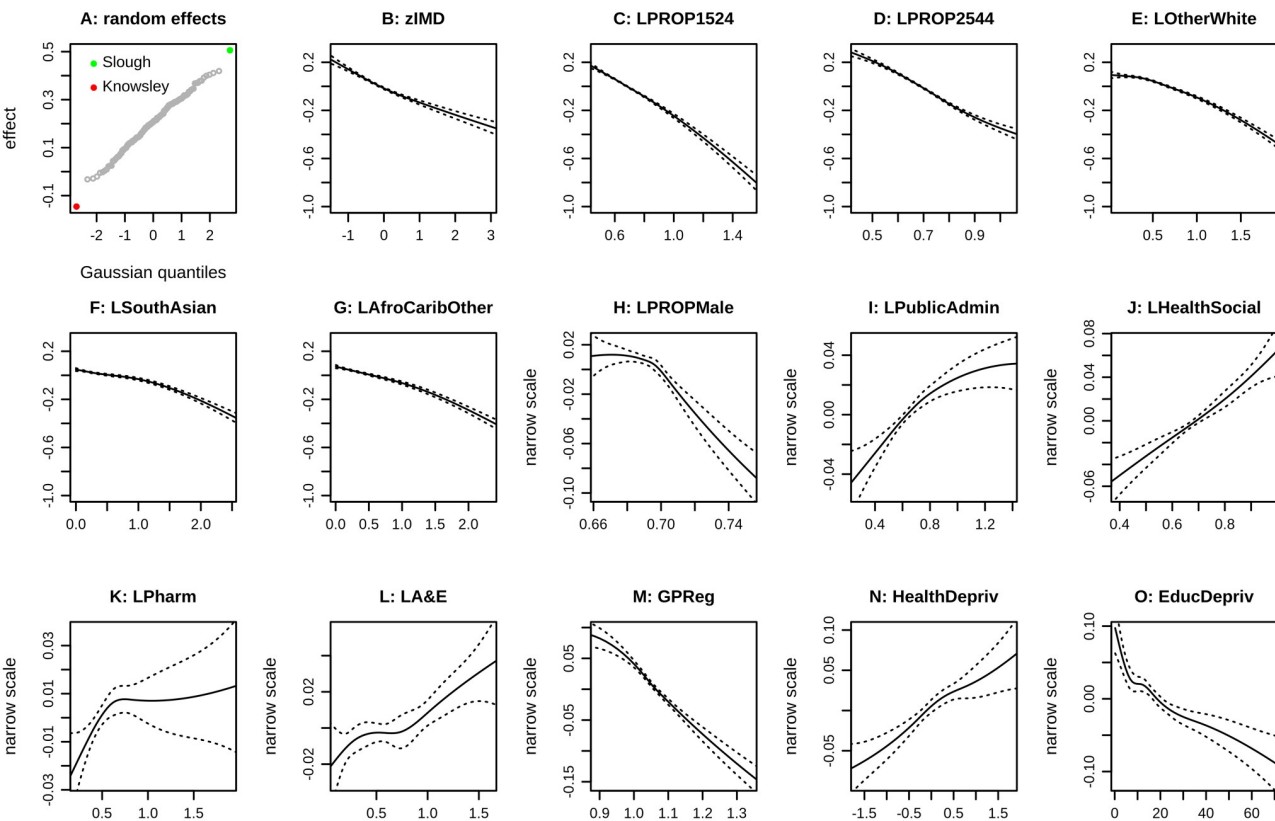

**Fig 2. Selected smoothers for model $\mathcal{D}$ for the Third Injection.** The panels show the modelled impact of individual covariates on the overall fitted value (displayed on the scale of the linear predictor, before it is translated to fitted output by the inverse link function). The x-axis is limited by the 0.01 and 0.99 quantiles of the covariate. In some panels the y-axis is limited by the range of the smoother and is labelled "narrow scale".

four models, and the North East is negative with model $\mathcal{D}$. London is a positive predictor with models $\mathcal{B}$ and $\mathcal{D}$.

A comparable table for the first dose shows that Bayes $R^2$ rises from 0.398 for modelling the estimated random effects from $\mathcal{A}$, to 0.464 for $\mathcal{B}$, 0.606 for $\mathcal{C}$, and 0.600 for $\mathcal{D}$. All four models lead to second stage models which show Flu vaccination and the rise in Public Health budgets as positive predictors of the estimated or simulated random effects, and likewise for London and South East regions, whilst North West is negative, and North East is negative for $\mathcal{B}$, $\mathcal{C}$, and $\mathcal{D}$. South West is positive for $\mathcal{C}$ and $\mathcal{D}$. Hospital hubs are positive for $\mathcal{C}$.

For the second dose, Bayes $R^2$ rises from 0.400 for modelling the estimated random effects from $\mathcal{A}$, to 0.462 for $\mathcal{B}$, 0.594 for $\mathcal{C}$, and 0.589 for $\mathcal{D}$. All four models lead to second stage models which show Flu vaccination and the rise in Public Health budgets as positive predictors of estimated or simulated random effects, and likewise for London and South East regions, whilst North West is negative, and North East is negative for $\mathcal{B}$, $\mathcal{C}$, and $\mathcal{D}$. South West is positive for $\mathcal{C}$ and $\mathcal{D}$, and borderline for $\mathcal{A}$.

Fig 5 shows the scaled estimated random effects from model $\mathcal{D}$ against the prediction from second stage modelling, for each vaccination.

## Discussion

There is extensive literature on COVID-19 vaccine uptake, including evidence from individuals. Before the vaccines were deployed, a telephone and web survey of attitudes, considered by

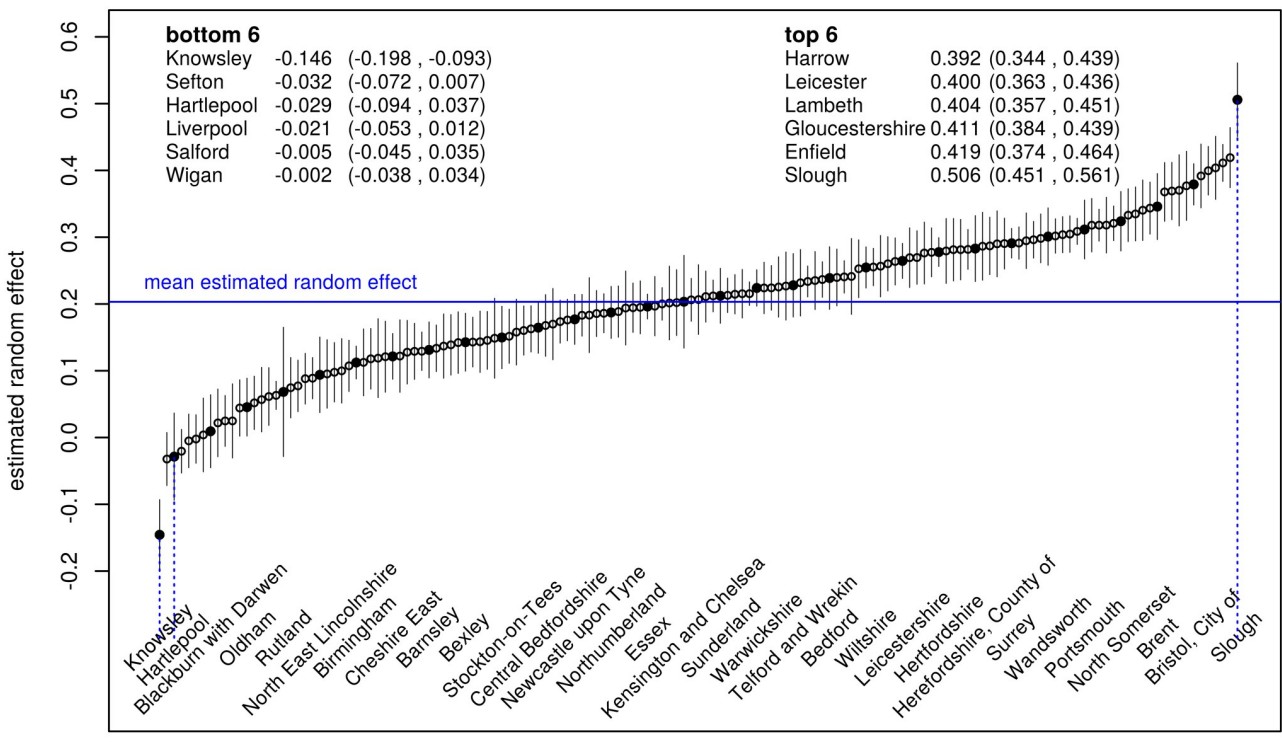

**Fig 3. "Caterpillar plot" for model $\mathcal{D}$ for the Third Injection.** Each estimated random effect is displayed with a point whose y-value is $c_i$ and a vertical segment showing its 95% credible interval. The points are ordered along the x-axis by increasing value of $c_i$. Points marked • also show the Upper Tier Local Authority name below. The blue line shows the mean value of $c_i$.

the UK government's advisory group SAGE in December 2020 found "marked differences existed by ethnicity, with Black ethnic groups the most likely to be COVID-19 vaccine hesitant followed by the Pakistani/Bangladeshi group. Other White ethnic groups (which includes Eastern European communities) also had higher levels of COVID-19 vaccine hesitancy than White UK/White Irish ethnicity" [3]. SAGE cited previous studies to conclude "Barriers to vaccine uptake include perception of risk, low confidence in the vaccine, distrust, access barriers, inconvenience, socio-demographic context and lack of endorsement, lack of vaccine offer or lack of communication from trusted providers and community leaders."

A UK cohort analysis by Helen Curtis et al [5] of 57.9 million patients' primary care records, concerns the period from 8 December 2020 to 17 March 2021. Of patients aged $\geq 80$ years not in a care home, 94.7% received a vaccine, but with substantial variation by ethnicity (White 96.2%, Black 68.3%) and deprivation (least deprived 96.6%, most deprived 90.7%).

A meta-analysis by Eric Robinson [8] of 28 nationally representative samples from 13 countries found "Being female, younger, of lower income or education level and belonging to an ethnic minority group were consistently associated with being less likely to intend to vaccinate". A meta-analysis by Qiang Wang [9] of international studies including representative samples found that "Gender, educational level, influenza vaccination history, and trust in the government were strong predictors of COVID-19 vaccination willingness" and that healthworkers were less willing than the general public.

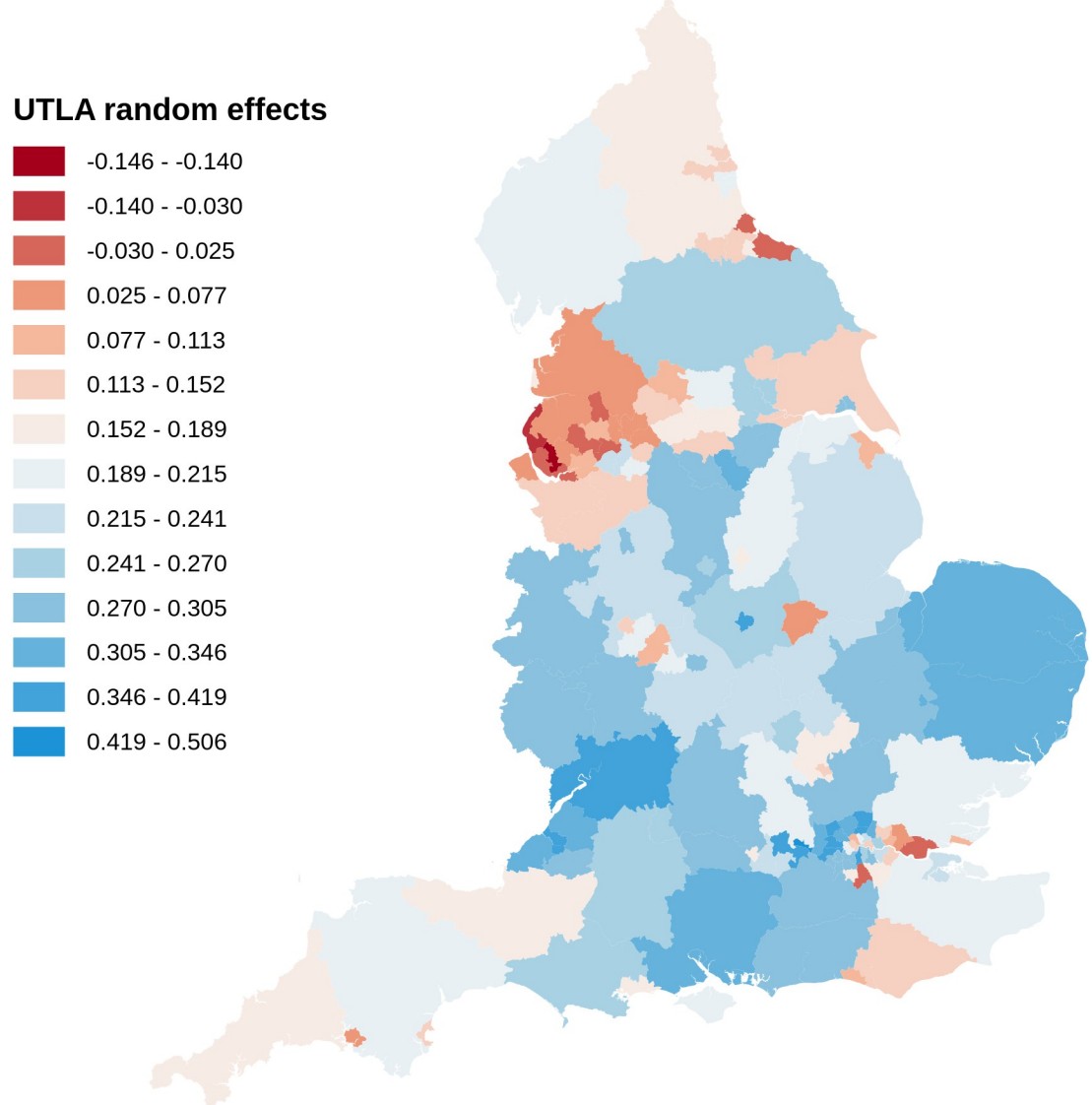

**Fig 4. Geography of estimated random effects of Upper Tier Local Authorities, for the Third Injection.** A map of England is shown with Upper Tier Local Authorities, each shaded according to the estimated random effect $c_i$ using model $\mathcal{D}$ for the Third Injection. Areas with red (blue) shading have estimated random effects below (above) the mean.

Recent flu vaccination history and trust in routine vaccination recommendations from health workers were amongst the positive factors in willingness-to-pay for COVID-19 vaccines in a survey by Malik Sallam et al [41] of individuals from 10 low-middle income countries in Asia, Africa, and South America.

An international review of COVID-19 vaccine hesitancy by Wardah Hassan et al [42] included a table of factors associated with acceptance or rejection in a wide range of countries, and highlighted the role of disinformation and negative religious beliefs as barriers.

Occupation also influences vaccine uptake. The Office for National Statistics Coronavirus (COVID-19) latest insights: Vaccines, (as of 31 December 2021) [43] shows survey data on how the proportion receiving 3 vaccinations varied by occupation groupings, ranging from

**Table 2. Parameter estimates from 2nd stage modelling of random effects for Third Injection.**

| | $\mathcal{A}$ | | $\mathcal{B}$ | | $\mathcal{C}$ | | $\mathcal{D}$ | |
|---|---|---|---|---|---|---|---|---|
| **Bayes $R^2$** | 0.203 | | 0.293 | | 0.353 | | 0.369 | |
| **$c_i$ estimated** | | | | | | | | |
| | estimate | 90% credible | | | | | | |
| Flu vaccination | 0.23 | (0.06,0.40)* | 0.23 | (0.06,0.40)* | 0.32 | (0.16,0.48)* | 0.29 | (0.13,0.45)* |
| Public Health change | 0.33 | (0.16,0.49)* | 0.22 | (0.06,0.38)* | 0.30 | (0.15,0.46)* | 0.21 | (0.06,0.37)* |
| Pharmacy vacc sites | -0.01 | (-0.15,0.14) | -0.06 | (-0.21,0.08) | -0.07 | (-0.21,0.07) | -0.06 | (-0.20,0.07) |
| GP vacc sites | 0.12 | (-0.02,0.25) | 0.06 | (-0.07,0.19) | 0.11 | (-0.01,0.23) | 0.08 | (-0.05,0.20) |
| Vaccination Centres | 0.12 | (-0.01,0.24) | 0.09 | (-0.04,0.21) | 0.13 | (0.01,0.25)* | 0.10 | (-0.02,0.22) |
| Hospital Vacc Hubs | 0.05 | (-0.08,0.18) | 0.02 | (-0.11,0.14) | 0.05 | (-0.08,0.17) | 0.04 | (-0.08,0.15) |
| East Midlands | 0.12 | (-0.18,0.53) | 0.26 | (-0.14,0.66) | 0.27 | (-0.11,0.66) | 0.27 | (-0.12,0.67) |
| East of England | -0.14 | (-0.53,0.15) | -0.03 | (-0.40,0.35) | 0.06 | (-0.30,0.44) | 0.10 | (-0.29,0.49) |
| London | 0.16 | (-0.10,0.49) | 0.43 | (0.09,0.79)* | 0.30 | (-0.00,0.62) | 0.47 | (0.14,0.80)* |
| North East | -0.01 | (-0.38,0.34) | -0.36 | (-0.85,0.07) | -0.31 | (-0.78,0.09) | -0.53 | (-0.99,-0.08)* |
| North West | -0.38 | (-0.75,-0.04)* | -0.79 | (-1.13,-0.47)* | -0.81 | (-1.12,-0.49)* | -0.94 | (-1.24,-0.63)* |
| South East | -0.08 | (-0.39,0.18) | 0.12 | (-0.19,0.42) | 0.16 | (-0.13,0.45) | 0.25 | (-0.05,0.55) |
| South West | 0.07 | (-0.21,0.42) | -0.02 | (-0.37,0.32) | 0.20 | (-0.13,0.54) | 0.14 | (-0.19,0.49) |
| West Midlands | 0.16 | (-0.11,0.53) | 0.28 | (-0.05,0.62) | 0.19 | (-0.14,0.53) | 0.23 | (-0.10,0.56) |
| Yorkshire & Humber | 0.10 | (-0.16,0.43) | 0.06 | (-0.27,0.38) | 0.04 | (-0.27,0.34) | -0.03 | (-0.36,0.28) |
| **$c_i$ simulated** | | | | | | | | |
| | estimate | 90% credible | | | | | | |
| Flu vaccination | 0.23 | (0.05,0.40)* | 0.23 | (0.05,0.41)* | 0.31 | (0.14,0.47)* | 0.28 | (0.11,0.45)* |
| Public Health change | 0.32 | (0.15,0.49)* | 0.22 | (0.04,0.39)* | 0.30 | (0.13,0.46)* | 0.21 | (0.04,0.37)* |
| Pharmacy vacc sites | -0.01 | (-0.15,0.14) | -0.05 | (-0.20,0.10) | -0.07 | (-0.21,0.08) | -0.06 | (-0.20,0.08) |
| GP vacc sites | 0.11 | (-0.03,0.25) | 0.06 | (-0.07,0.19) | 0.11 | (-0.02,0.23) | 0.08 | (-0.05,0.20) |
| Vaccination Centres | 0.11 | (-0.02,0.25) | 0.08 | (-0.05,0.21) | 0.13 | (0.00,0.25)* | 0.10 | (-0.03,0.22) |
| Hospital Vacc Hubs | 0.04 | (-0.10,0.18) | 0.02 | (-0.11,0.16) | 0.04 | (-0.09,0.17) | 0.04 | (-0.09,0.16) |
| East Midlands | 0.14 | (-0.18,0.52) | 0.26 | (-0.14,0.68) | 0.27 | (-0.11,0.68) | 0.27 | (-0.13,0.68) |
| East of England | -0.15 | (-0.52,0.16) | -0.03 | (-0.43,0.36) | 0.05 | (-0.32,0.43) | 0.10 | (-0.29,0.49) |
| London | 0.16 | (-0.11,0.49) | 0.42 | (0.06,0.80)* | 0.30 | (-0.03,0.64) | 0.46 | (0.11,0.81)* |
| North East | -0.02 | (-0.38,0.33) | -0.37 | (-0.85,0.08) | -0.31 | (-0.76,0.11) | -0.51 | (-0.99,-0.05)* |
| North West | -0.36 | (-0.73,-0.01)* | -0.76 | (-1.11,-0.41)* | -0.78 | (-1.11,-0.44)* | -0.91 | (-1.23,-0.59)* |
| South East | -0.09 | (-0.38,0.18) | 0.11 | (-0.20,0.44) | 0.16 | (-0.14,0.47) | 0.24 | (-0.07,0.56) |
| South West | 0.08 | (-0.22,0.40) | -0.02 | (-0.38,0.33) | 0.19 | (-0.15,0.54) | 0.13 | (-0.22,0.49) |
| West Midlands | 0.18 | (-0.11,0.52) | 0.27 | (-0.07,0.63) | 0.19 | (-0.14,0.53) | 0.21 | (-0.13,0.56) |
| Yorkshire & Humber | 0.11 | (-0.17,0.43) | 0.06 | (-0.28,0.40) | 0.02 | (-0.30,0.35) | -0.03 | (-0.36,0.31) |

The first line shows the Bayes $R^2$ (see Statistical method) when second stage modelling is applied to the scaled estimated random effects for each of four models applied to the Third Injection. Below this, the upper portion shows the parameter estimates and 90% credible intervals for each covariate in the second stage model. The lower portion uses second stage modelling of scaled simulated random effects (see Statistical method). Credible intervals which exclude 0 are starred.

80.4% (for persons employed in Health) to 48.0% (Elementary trades and related occupations); and for more specific occupations; and by ethnicity, ranging from 68.4% (White British) to 33.9% (Black Caribbean) [44].

Vaccination status is an important predictor of mortality, though some ethnic contrasts persist even when controlling for it. An ONS study of ethnic contrasts in coronavirus death rates 8 December 2020 to 1 December 2021 found that "Location, measures of disadvantage, occupation, living arrangements, pre-existing health conditions and vaccination status

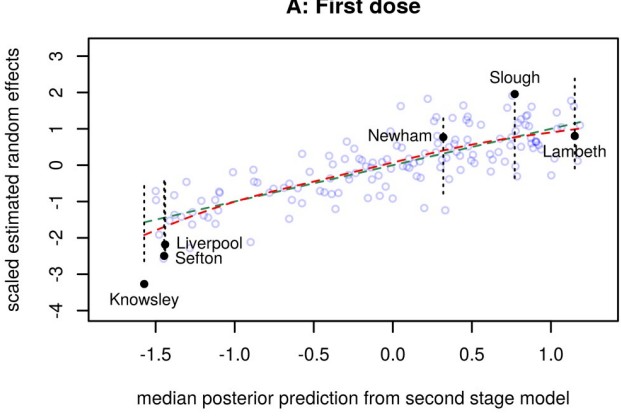
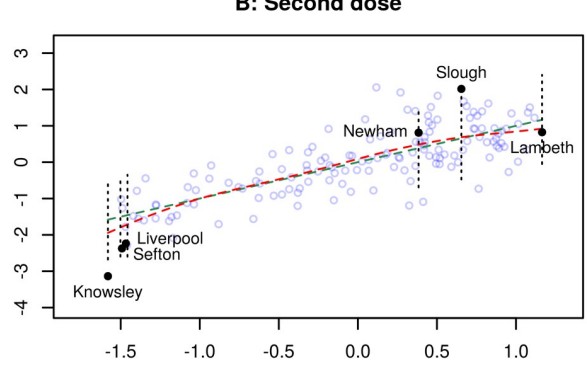

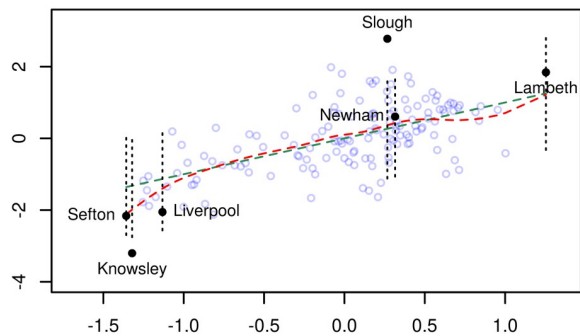

**Fig 5. Second stage prediction of scaled estimated random effects from model** $\mathcal{D}$**.** For each vaccination, the corresponding panel shows the scaled estimated random effects $c_i$ against the median posterior predictive values from the second stage model. The dashed green line shows the predictions, and the dashed red line shows a smooth spline through the $c_i$. Points marked • are selected Upper Tier Local Authorities mentioned elsewhere in the text. For these points, the vertical dotted lines show the 90% credible interval of posterior predictive values.

accounted for a large proportion of the excess rate of death involving COVID-19 in most ethnic minority groups; however, the Bangladeshi ethnic group and men from the Pakistani ethnic group remained at higher risk than White British people in the third wave, even after adjusting for vaccination status" [45].

Several authors have made recommendations based on the impact of individual characteristics. Mohammed S Razai outlined a strategy to overcome vaccine hesitancy [6], highlighting Confidence (importance, safety and efficacy of vaccines); Complacency (perception of low risk and low disease severity); Convenience (access issues dependent on the context, time and specific vaccine being offered); Communications (sources of information); and Context (sociodemographic characteristics). In an article entitled "What must be done to tackle vaccine hesitancy and barriers to COVID-19 vaccination in migrants?" [7] Alison Crawshaw highlighted "mistrust of the state and health system, stemming from historical events, data sharing policies and dissatisfaction with the initial handling of the pandemic" and advocated "engaging with communities to understand their concerns or barriers to vaccination and working together to co-develop tailored approaches to encourage uptake and rebuild trust".

## Geographical factors

Individual characteristics are not the only factors influencing COVID-19 vaccine uptake. An ONS technical article [46] considered vaccination status of individuals, controlling for sex,

ethnicity, age, geographical region, urban or rural classification of their address, deprivation percentile, household size, whether the household was multigenerational. The model outputs [47] include regional variation, with negative impact of residence in the North East, North West, and West Midlands amongst persons aged 18–34.

Geographical factors are also known to influence the uptake of childhood vaccinations. A Michigan study by ST Hegde of Neighborhood Influence on Diphtheria-Tetanus-Pertussis Booster Vaccination [48] considered individual data on uptake along with covariates at the level of US census tract (average 4,000 people) and the much smaller units of census block, whose socio-economic characteristics were taken as proxies for the individual household. Tract level affluence and socio-economic disadvantage both showed negative impact on child vaccination uptake.

A study of childhood Measles, Mumps, Rubella (MMR) immunisation rates in Italy by Veronica Toffolutti [49], using annual data at regional level from 2000 to 2014, found that reductions in public health budgets were a significant predictor of falling MMR rates. The declines after 2010 were concentrated in those regions which experienced the greatest reductions in the real public health expenditure. Modelling found that each 1% reduction in real public health expenditure per-capita corresponded to an 0.5% reduction in MMR coverage. However, the models do not appear to control for socio-economic variation between regions.

In the geographical analysis reported here, I initially sought to learn whether the current very low booster uptake in particular small areas of England was unexpected, or could be predicted from the national data on the UK Coronavirus Dashboard, at Middle Super Output level. The observations are the grouped data for each MSOA, and as such they contain no direct information on the individuals within the area, unlike many of the studies cited above. However, the MSOA data is comprehensive, rather than a sample. Planning relies on area data, and so it is relevant to know which characteristics at area level are associated with variation in vaccination uptake, which will in turn lead to variation in rates of serious infection and demand for healthcare. Although associations at area level do not imply the same associations for individuals, they are at least suggestive of individual impacts to be examined, and can be compared with results based on individual data.

In each MSOA, the data shows the number of people eligible for the vaccine, and the cumulative number actually vaccinated with the first and second doses, and the Third Injection. The latter covers both people receiving the booster, and those with compromised immunity who were given a third primary dose. The ratio of those vaccinated to those eligible, is the uptake. But the actual numbers vaccinated and eligible give more information, and can be modelled as a binomial variable.

The Generalized Additive Model, implemented in R with the mgcv package, is an established technique for non-linear modelling. The variance in this data exceeds what would be expected for a binomial variable, but can be estimated by relaxing the "binomial" assumption to "quasibinomial". Using demographic, employment and local health data at MSOA-level as predictors, such models fit this data well. They are improved by including a random effect assigned to the Upper Tier Local Authority within which the MSOA is found, allowing other aspects of the UTLA to influence the prediction. This is analogous to the neighborhood influence on individual uptake of the DTP booster [48], but with MSOAs rather than individuals as the unit of analysis, and grouped by UTLAs rather than census tracts.

Four different models, for each of the three vaccinations, all gave a good fit to the data with the proportion of deviance explained ranging from 95.3% to 97.2%. Fig 1 shows how well the preferred model $\mathcal{D}$ performs for the Third Injection. This close fit means that the observed values in a particular MSOA are almost always near the prediction from the national data. Including the random effects term significantly improves the model fit.

The covariate structure was simplified to enable models to be computable on a PC. For example, whilst population data is available for each year of age, the models use broader categories such as aged 15–24 or 25–44. Even so, $\mathcal{D}$ has 33 smoothers, each with a smoothing parameter which is estimated during fitting, along with 341 coefficients.

The smoothers for $\mathcal{D}$ for the Third Injection, some of which are shown in Fig 2, show the impact of specific covariates in the context of all others in the model. The most powerful fixed effects are the Index of Multiple Deprivation (Fig 2B), the population proportions aged 15–24 (Fig 2C), aged 25–44 (Fig 2D), Other White ethnicity (Fig 2E), South Asian (Indian, Pakistani and Bangladeshi combined) (Fig 2F), and the combined group of African, Caribbean, Other Black, and Other ethnicity (Fig 2G).

The negative slope of the smoother in Fig 2B shows that higher IMD leads to a lower predicted uptake if all other variables are unchanged. Education Deprivation (a specific domain within the overall index IMD) also has a negative slope (Fig 2O), although its impact is smaller and is shown with "narrow scale". All six most powerful fixed effects have smoothers with negative slope. The dependence on age is to be expected, as the vaccine rollout targeted different age groups at different times, but Aged 15–24 is particularly negative (Fig 2C).

Ethnicity has significant impact on uptake at MSOA level, as many other studies have found for individuals. The smoother for the "Other White" group (Fig 2E) has the widest range, comparable to that for IMD (Fig 2B), showing that the "Other White" proportion of the population has a strong impact on predicted uptake.

Several other smoothers have small positive slope, such as those for the proportion employed in Public Adminstration (Fig 2I), or Health and Social Care (Fig 2J). The latter indicates that despite the well publicised vaccine hesitancy amongst a minority of NHS and care staff [50], and the international meta-analysis [9], the overall impact of employment in these sectors in England is to increase the area uptake of the Third Injection, as in the ONS survey data [43]. For the first and second doses, the smoother rises when the proportion of Health and Care staff exceeds the 20th percentile of this covariate.

The smoother for average distance to A&E (Fig 2L) also has a positive slope, suggesting that people living further from an emergency department may be more concerned to take the vaccination. Likewise, increasing Health Deprivation (Fig 2N) is associated with increasing uptake. Conversely, increased GP registration (Fig 2M) is associated with decreasing uptake. However, there is some uncertainty in the GP registration data, and NHS Digital have commented on the excess of GP registrations over ONS population estimates [51].

The smoother for male (Fig 2H) shows uptake of the Third Injection declining with increasing male proportion of the population. This differs from the international meta-analyses [8, 9], and from the ONS technical report [47]. However, the meta-analyses concern data from 2020, and the ONS report does not show a significant negative uptake for females during time periods after 5 September 2021, which cover the Third Injection. Model $\mathcal{D}$ includes many more covariates including employment sectors, which may affect the smoother for male. In any case, whilst the smoother is highly significant (p<2e-16), its range is narrow as shown in Fig 2H.

Fitting any of these models also gives estimates for the estimated "random effect" of each of the 149 UTLA. For example "Knowsley" has a strong negative impact, and "Slough" has a strong positive impact (as shown for model $\mathcal{D}$ in Fig 2A). These random effects are not to be confused with the actual uptake of vaccination in the local authority, which may be low or high due to deprivation or other fixed effects, and which may vary widely amongst MSOAs within the local authority. Uptake of the Third Injection varied from 32.2% to 57.6% within Knowsley, from 17.3% to 65.5% in Liverpool, and from 36.9% to 75.8% in Sefton.

The random effect alters the prediction from the fixed effects alone. If Beckton Park (in Newham) had been located in Slough with identical demography, employment, and health

indicators, the predicted uptake of the Third Injection would rise by 19%, whilst if it were in Knowsley the prediction would fall by 28%.

The range in magnitude of the random effects is comparable to that of ethnicity groups. On the scale of the linear predictor, the smoothers for Other White, South Asian, and African-Caribbean-Other groups range from 0.09 to -0.70, 0.05 to -0.42, and 0.07 to -0.51 respectively, each smoother descending as the ethnicity proportion rises. The random effects range from -0.15 (Knowsley) to 0.51 (Slough). The difference between the random effects for Knowsley and Slough is greater than the maximal difference between any two MSOA due to the population proportion of South Asian or African-Caribbean-Other ethnicity, and only slightly less than the maximal difference due to the proportion of Other White.

Fig 3 shows that many UTLA have estimated random effects whose 95% credible interval does not include the mean value. Thus, as well as the entire random effects term being highly significant, particular UTLAs can be distinguished from each other and from the mean value.

As Fig 4 shows, there is a strong Regional component to the variation in random effects, with the North West containing many low values. The second stage of modelling sought to explain the estimated random effects in terms of other information at UTLA level. This stage tested the impact of flu vaccination rates, change in public health budgets, number of vaccination sites (per UTLA population) of four types, and Region, treated as a random factor. Simulated random effects generated from the first stage model were used to refine the parameter estimates and credible intervals from the second stage, but this made little difference. As Table 2 shows, the parameters and credible intervals for second stage modelling of random effects estimated from model $\mathcal{D}$ are similar to those after the random effects are simulated, using the corrected covariance matrix from $\mathcal{D}$. This method is illustrated with synthetic data in the code file simutest1 (see S2 File), described in the Guide (S1 File).

The second stage model passes Bayesian checks and shows clear impacts of Region, flu vaccination rates, and change in public health budgets. Fig 5 shows the extent to which the chosen predictors actually account for the estimated random effects. None of the points are outliers (by Pareto k) and the model fits well for the first and second doses with Bayes $R^2 \sim 0.6$. The lower value for the Third Injection, $\sim 0.37$, suggests that there may be other relevant covariates at UTLA level.

It is striking that Knowsley, Liverpool and Sefton all appear at the lower left of each panel in Fig 5 and Knowsley is conspicuously low for all three vaccinations. The Merseyside local authorities are amongst the most deprived in England, but IMD already appears in the first stage model $\mathcal{D}$ as a fixed effect so was not expected to have any impact on the second stage model. Indeed, if IMD is averaged over the MSOA within each UTLA and then used as a predictor in second stage modelling, its 90% credible interval includes 0.

Simply being in the North West is the most powerful predictor of low uptake in this model, and in addition Knowsley had the 13th lowest increase in public health funding (and the third lowest within the North West), rising by only 0.88%. Newham benefits from being in London and from the larger increase in its public health budget (rising by 2.09%), and possibly from the vaccination centre located in Olympic Park, an outlier with positive residual (see Fig 1). Flu vaccination rates are higher in Knowsley (49.2) than in Newham (45.5), so cannot explain the disparity in random effects.

Slough benefits from being in the South East and possibly from a vaccination centre, whilst its flu vaccination rate and rise in public health budget are both close to their respective mean values. However the random effects for Slough also exceed prediction. The highest predicted value is in Lambeth, where the flu vaccination rate is below average, but which benefits from being in London with a 4.88% rise in public health budget and 3 vaccination centres. Its estimated random effect is close to prediction for each vaccination.

All such conclusions depend on the validity of the models. The first stage MSOA level modelling fits very well, and the resulting estimated random effects are similar from all four models. That is, the random effects are not simply artefacts of the model, once ethnicity is included, although they change dramatically if it is omitted. Correlations between estimated random effects when fitting the same model to different vaccinations show that UTLAs which performed better (worse) for the first two doses, generally did so for the Third Injection.

This analysis concerns data as of 4 January 2022, and cannot show how the performance of UTLAs changed over time. There may be other relevant covariates at UTLA or MSOA level not included here. For example, the Understanding Society: COVID-19 Study, 2020–2021 [52] may contain relevant covariates which can be linked to geographic information [53]. Social media use could indicate the local influence of disinformation. Data on ethnicity and occupation was taken from the 2011 Census and may be outdated, though it was the most recent available. Ideally all of the UTLA and MSOA level variables would be incorporated in a single Bayesian model. But that appeared prohibitively slow to compute, and the relevant programme gamm4 used with rstanarm does not admit quasi families, so the random effects estimates from the first stage GAM modelling were considered as observations, to be modelled in their own right and then refined by simulation. The first stage modelling used the quasibinomial family, and other possibilities were not investigated. A random intercept for UTLA was the simplest possible random effect structure, but UTLA may also interact with some of the fixed covariates. Likewise, all of the fixed effects were given independent smoothers, without interactions. However, the chosen model did give a good fit to the data.

## Conclusion

The UTLA level random effects could be described as a postcode lottery, as they are not explained by the population characteristics controlled for by the fixed effects, but are associated with other geographical factors. However, a "lottery" suggests pure chance whereas Regional disparity and changes in public health budgets impact on COVID-19 vaccination rates. In turn, these factors reflect economic policy decisions.

Much of the literature has focused on "vaccine hesitancy" of specific population subgroups at individual level. The modelling here confirms the impact on area uptake, of ethnicity along with deprivation and age, but also identifies additional factors at MSOA level and others characteristic of the wider locality rather than the population living in it, as with evidence on childhood immunisations in Italy.

These models indicate that whatever other barriers exist due to deprivation and within particular ethnicities, the annual change in local authority public health budgets is also a positive predictor. Therefore, increasing local public health allocations would be one simple way to improve COVID-19 vaccine uptake.

## Supporting information

**S1 File. Guide.** Guide to the files contained within the zip files for codes, data, and outputs, how to run the codes, and short simulations to illustrate second stage modelling.
(PDF)

**S2 File. Codes.** A zip file containing the codes used to model the data.
(ZIP)

**S3 File. Data1.** A zip file containing most of the data files.
(ZIP)

**S4 File. Data2.** A zip file containing large data files for GP registration and IMD.
(ZIP)

**S5 File. Data3.** A zip file containing large data files for MSOA vaccinations and AHAH Output.
(ZIP)

**S6 File. Outputs.** A zip file containing the outputs of modelling. This can be used to load the fitted models.
(ZIP)

**S7 File. Summary.** Summary of model $\mathcal{D}$ applied to the Third Injection.
(PDF)

**S1 Appendix. Appendix.** Brief outline of how GAM models are fitted, smoothing parameter optimisation by "Marginal Likelihood", and the standard errors of fitted coefficients.
(PDF)

## Acknowledgments

Thanks to David Taylor Robinson for helpful comments including the AHAH dataset and the suggestion that budgets could be a predictor, and to Isabelle Whelan for an unpublished essay "Austerity, NHS reform and vaccine uptake", written before the pandemic. Thanks also to the academic editor Harapan Harapan and both anonymous reviewers for comments during peer review.

## Author Contributions

**Conceptualization:** Greg Dropkin.

**Formal analysis:** Greg Dropkin.

**Writing – original draft:** Greg Dropkin.

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
