## [Decision Letter · Decision Letter 0]

3 May 2022

PONE-D-22-05078

Variation in English Covid booster uptake: Generalized Additive Model

PLOS ONE

Dear Dr. Dropkin,

Thank you for submitting your manuscript to PLOS ONE. After careful consideration, we feel that it has merit but does not fully meet PLOS ONE’s publication criteria as it currently stands. Therefore, we invite you to submit a revised version of the manuscript that addresses the points raised during the review process.

We look forward to receiving your revised manuscript.

Kind regards,

Harapan Harapan, MD, PhD

Academic Editor

PLOS ONE

2.Please amend either the abstract on the online submission form (via Edit Submission) or the abstract in the manuscript so that they are identical.

5. We note that [Figure 4] in your submission contain [map/satellite] images which may be copyrighted. All PLOS content is published under the Creative Commons Attribution License (CC BY 4.0), which means that the manuscript, images, and Supporting Information files will be freely available online, and any third party is permitted to access, download, copy, distribute, and use these materials in any way, even commercially, with proper attribution. For these reasons, we cannot publish previously copyrighted maps or satellite images created using proprietary data, such as Google software (Google Maps, Street View, and Earth). For more information, see our copyright guidelines: http://journals.plos.org/plosone/s/licenses-and-copyright.

a. You may seek permission from the original copyright holder of Figure(s) [#] to publish the content specifically under the CC BY 4.0 license. 

Natural Earth (public domain): http://www.naturalearthdata.com/.

6. Please update your submission to use the PLOS LaTeX template. The template and more information on our requirements for LaTeX submissions can be found at http://journals.plos.org/plosone/s/latex.

Reviewers' comments:

Reviewer's Responses to Questions

**Comments to the Author**

1. Is the manuscript technically sound, and do the data support the conclusions?

Reviewer #1: Yes

Reviewer #2: Partly

2. Has the statistical analysis been performed appropriately and rigorously? 

Reviewer #1: Yes

Reviewer #2: I Don't Know

3. Have the authors made all data underlying the findings in their manuscript fully available?

Reviewer #1: Yes

Reviewer #2: No

4. Is the manuscript presented in an intelligible fashion and written in standard English?

Reviewer #1: Yes

Reviewer #2: Yes

5. Review Comments to the Author

Reviewer #1: in the abstract results, what do you mean by 90% significance? Note - I think you use Bayesian techniques and therefore refer to 90% credibility intervals throughout. But I would stay away from making inference about "significance" within a Bayesian framework.

some sort of figure or flowchart might be useful to explain or diagram what middle super output areas versus upper tier local authorities are. An explanation within the text of what these mean (in terms of average population and area) also could be useful.

The other uses publicly available data about area-level variables on area-level vaccination. But some of the inferences are for an individual level. Distinguishing whether the author thinks of these as a proxxy for an individual level or if he is just studying area-level associations as an ecological association.

On that note, I personally would appreciate some connection to the existing literature on multilevel or neighborhood effects (examples similar but not limited to:

Diez-Roux A. Multilevel analysis in public health research. Annu Rev Public Health. 2000;21: 171–192.

Hegde ST, Wagner AL, Clarke PJ, Potter RC, Swanson RG, Boulton ML. Neighborhood Influence on Diphtheria-Tetanus-Pertussis Booster Vaccination. Public Health. 2019;167: 41–49. https://doi.org/10.1016/j.puhe.2018.11.009

I think the GAM is not the important part that I would emphasize in the title - maybe something like: COVID booster uptake in England: an ecological study

When did booster / 3rd dose vaccination start in England? Would be useful to include this in Intro or Methods.

I do appreciate the attention to detail in the methods. I think as is would be fine, but my recommendation would be to put much of what you have in the appendix along with your guide, and instead explain it in more vague terms which are not specific to the coding in R. Again - this coding is incredibly useful for people to replicate the results, but to me that would be more appropriate for the supplementary appendix.

I would stay away from using causal language like "effects" and instead use terms like association, or maybe even impact, since you have an observational study.

Subheadings in the discussion could be useful. It seems like you have one section related to relating the results to previous literature, one section related to explaining geographic variation with less comments - that could be helpful as an initial structure. As is it is hard to follow your thought process across paragraphs.

Reviewer #2: The author performed Generalized Additive Model analyses on the Covid vaccination data from the UK Government Coronavirus Dashboard. In general, the writing requires heavy improvement, where many technical errors (such as logical flows of the paragraph, captions of charts and tables, data presentation) could be found here and there. I have some recommendations to guide author in revising their manuscript.

SUPPLEMENTARY FILE

1. I could not find/access the supplementary or supporting files author stated in the manuscript. Probably making a special section about Data Availability at the end of the manuscript would help.

ABSTRACT

2. “Are there other relevant predictors?” So, what are other predictors? Please state in the last part of the abstract.

3. What is MSOA? All abbreviation should be defined first.

4. The “discussion” part fits better with “conclusions”

INTRODUCTION

5. “Vaccination uptake is highly dependent on age, gender, ethnicity, and deprivation, as is widely recognised 3 4 5 6 7 8 9” Are all the studies cited from UK? Or, are they from different countries? Readers may assume they are all from UK, I suggest author to breakdown each of the cited literatures.

6. “…in contrast with 41.1% in Liverpool and 67.4% in

Gloucestershire.” Where does this information come from? Please cite the literature.

7. Please define abbreviations: MSOA, UTLA, etc.

8. In the last paragraph, author is encouraged to elaborate as to why this approach is appropriate. Is it more robust in comparison with other approaches?

METHODS

9. “Data sources” and “Statistical method and models” should be under one section “Methods”

RESULTS

10. Table 1 has no border, and should be revised. Not sure what this value mean -15317.25 (0.953), I understand that author put it in the caption, but that is not enough. Furthermore, why author does not make it uniform, e.g. Dose 1, Dose 2, Dose 3 or 1st injection, 2nd injection, 3rd injection?

11. Before Table 1, the paragraph only consists of 1 sentence. I encourage the author to move more sentences into the paragraph. And this applies for every data presented (including the following charts or tables).

12. “….except for distance to pharmacy (p<0.01).” Where can I see the data? Supplementary file? Author should cite it in the text.

DISCUSSION

13. “…COVID-19 vaccine hesitant followed by the Pakistani/Bangladeshi group.” Provide citation please.

14. In the second paragraph of Discussion, consider to include these following studies.

# In a study performing willingness-to-pay analysis, flu vaccination is also among the positive predictors for COVID-19 vaccine acceptance. Citing: Sallam et al. Narra J 2022; 2(1): e74 – doi: 10.52225/narra.v2i1.74

# Side effects of COVID-19 vaccine become the major barrier for vaccination program in population from different countries. Citing: Rosiello et al. Narra J 2021; 1(3): e55- doi: 10.52225/narra.v1i3.55.

#Disinformation and certain religious narrative could discourage individuals to take COVID-19 vaccine. Citing: Hassan et al. Narra J 2021; 1(3): e 57 - doi: 10.52225/narra.v1i3.57

15. I don’t understand what is the author trying to convey in the fourth paragph of the Discussion.

16. This paragraph, “An ONS technical article…” Does author try to compare the method with previous study? I think this paragraph better fit in the introduction, rather than in discussion. Also, clarify the intention of putting this paragraph, because it seems author just arbitrarily put random paragraph in the Discussion.

17. Discussion needs a lot of re-arrangements to make it more readable and acceptable for publication. Other than re-arranging the paragraphs to optimize the logical flow, author is encourage to distinct the findings from previous studies and that of from present study, and followed by a clear comparison. At the present version, it’s very hard to get the comparative pictures of the studies.

18. Please provide limitation of the study at the end of discussion.

19. Please provide a section of “Conclusions”

6. PLOS authors have the option to publish the peer review history of their article (what does this mean?). If published, this will include your full peer review and any attached files.

Reviewer #1: No

Reviewer #2: No

---

## [Author Response · Author response to Decision Letter 0]

4 Jun 2022

Thanks for all these comments, which I have tried to deal with in the amended version.

Journal requirements

1. Response: I have tried to comply with the style requirements including file names.

2. Response: I have amended the Abstract on the online submission to agree with the revised manuscript.

3. Response: Each figure and table now has a caption.

4. Response: The Supporting Information files have captions and are cited accordingly.

5. Response: Figure 4 is not a reproduction, but is an original map whose shading is derived from the modelling in this paper. To my knowledge it has not been published elsewhere, and I am unaware of any equivalent published modelling from which this map could be derived. The upper tier local authority boundaries which enclose the areas separately shaded in the map, are freely available for download from the UK Office for National Statistics, and I have amended the text and endnotes to comply with the conditions for their use, which involves no other copyright. 

6. Response: I have used LaTex for the revision.

Reviewers responses to Questions

Reviewer #1: in the abstract results, what do you mean by 90% significance? Note - I think you use Bayesian techniques and therefore refer to 90% credibility intervals throughout. But I would stay away from making inference about "significance" within a Bayesian framework.

Response: Thanks, almost everything now refers to credible intervals, although those in the first stage models (and in Figure 2) are 95% credible. The first stage modelling is not Bayesian, but the intervals are now called “credible” because they are derived from standard errors obtained from the posterior covariance as now outlined in S1_Appendix. The second stage modelling is Bayesian and I report 90% credible intervals. However, the p-values computed by “mgcv” in the first stage modelling are approximate tests of significance, as they represent the approximate probability that a smooth term is actually 0. I do think it is appropriate to highlight those smooth terms which are extremely significant (p<2e-16) according to this test. I have cited the reference below Wood SN (2013b) as well as Wood’s book. 

From the mgcv help line:

P-values for smooth terms are usually based on a test statistic motivated by an extension of Nychka's (1988) analysis of the frequentist properties of Bayesian confidence intervals for smooths (Marra and Wood, 2012). These have better frequentist performance (in terms of power and distribution under the null) than the alternative strictly frequentist approximation. When the Bayesian intervals have good across the function properties then the p-values have close to the correct null distribution and reasonable power (but there are no optimality results for the power). Full details are in Wood (2013b), although what is computed is actually a slight variant in which the components of the test statistic are weighted by the iterative fitting weights. 

Nychka (1988) Bayesian Confidence Intervals for Smoothing Splines. Journal of the American Statistical Association 83:1134-1143.

Marra, G and S.N. Wood (2012) Coverage Properties of Confidence Intervals for Generalized Additive Model Components. Scandinavian Journal of Statistics, 39(1), 53-74.

Wood, S.N. (2013b) On p-values for smooth components of an extended generalized additive model. Biometrika 100:221-228

Reviewer #1: some sort of figure or flowchart might be useful to explain or diagram what middle super output areas versus upper tier local authorities are. An explanation within the text of what these mean (in terms of average population and area) also could be useful.

Response: I hope that the text is now clear, without a figure, but including the average MSOA population, the total number of MSOA, the fact that each MSOA belongs to a unique UTLA, the total numbers of MSOA and UTLA and the range of numbers of MSOA contained in a UTLA. 

Reviewer #1: The other uses publicly available data about area-level variables on area-level vaccination. But some of the inferences are for an individual level. Distinguishing whether the author thinks of these as a proxxy for an individual level or if he is just studying area-level associations as an ecological association.

Response: I am just studying area-level associations, but I point out in the discussion that several of these area-level associations are similar to individual survey results from other papers. None of the inferences in this paper are for an individual level and I am not regarding MSOA-level covariates as a proxy for individual values.

Reviewer #1: On that note, I personally would appreciate some connection to the existing literature on multilevel or neighborhood effects (examples similar but not limited to:

Diez-Roux A. Multilevel analysis in public health research. Annu Rev Public Health. 2000;21: 171–192.

Hegde ST, Wagner AL, Clarke PJ, Potter RC, Swanson RG, Boulton ML. Neighborhood Influence on Diphtheria-Tetanus-Pertussis Booster Vaccination. Public Health. 2019;167: 41–49. https://doi.org/10.1016/j.puhe.2018.11.009

Response: Thanks, and I have mentioned them both and clarified that the random effects in this paper are “random intercept”, the simplest possible type, although they are embedded in GAMs which are more complex than the linear models in the Diez-Roux review. I have also cited another review of linear and generalized linear mixed models and a book which includes non-linear mixed models, though not using GAMs. The Hegde paper does not display the model explicitly, so I am unclear if it is linear or nonlinear in the covariates at block level which are taken as proxies for individual level, but I cite it in the discussion as it shows area (tract level) impacts on vaccination uptake.

Reviewer #1: I think the GAM is not the important part that I would emphasize in the title - maybe something like: COVID booster uptake in England: an ecological study

Response: Agree, thanks.

Reviewer #1: When did booster / 3rd dose vaccination start in England? Would be useful to include this in Intro or Methods.

Response: Thanks, now included in the Intro.

Reviewer #1: I do appreciate the attention to detail in the methods. I think as is would be fine, but my recommendation would be to put much of what you have in the appendix along with your guide, and instead explain it in more vague terms which are not specific to the coding in R. Again - this coding is incredibly useful for people to replicate the results, but to me that would be more appropriate for the supplementary appendix.

Response: Thanks, I’ve deleted any mention of R codes from the “Statistical methods” section, which I hope is now the minimum needed to see what the first and second stage models are doing. There is now a short S1_Appendix.pdf which also omits R codes but gives a brief mathematical overview. It also explains some fitted model outputs including “ML” and “Proportion of Deviance Explained” which are used in Table 1, and the standard errors used in making Figures 2 and 3. S1_File.pdf is a Guide which points to the code files. 

Reviewer #1: I would stay away from using causal language like "effects" and instead use terms like association, or maybe even impact, since you have an observational study.

Response: Thanks, I am now only using “effects” in the phrases “random effects” and “fixed effects” which I think are the correct terms in these models. Elsewhere, I’ve generally used impact rather than association.

Reviewer #1: Subheadings in the discussion could be useful. It seems like you have one section related to relating the results to previous literature, one section related to explaining geographic variation with less comments - that could be helpful as an initial structure. As is it is hard to follow your thought process across paragraphs.

Response: Thanks, I hope it is clearer now with the rewrites, some reordering, and a subhead (“Geographical factors”) in the discussion.

Reviewer #2: The author performed Generalized Additive Model analyses on the Covid vaccination data from the UK Government Coronavirus Dashboard. In general, the writing requires heavy improvement, where many technical errors (such as logical flows of the paragraph, captions of charts and tables, data presentation) could be found here and there. I have some recommendations to guide author in revising their manuscript.

Response: Thanks, I hope the writing is clearer now.

Reviewer #2: SUPPLEMENTARY FILE

1. I could not find/access the supplementary or supporting files author stated in the manuscript. Probably making a special section about Data Availability at the end of the manuscript would help.

Response: The Supporting information section at the end of the revised manuscript now lists 7 files and an appendix. S1_File.pdf is a Guide with details of all the files, including codes, data, and output. It also explains how to run the code files, and outlines the simulations which illustrate the modelling technique using synthetic data. There also a summary of model D applied to the Third Injection, and a separate Appendix which outlines the method of fitting the models.

Reviewer #2: ABSTRACT

2. “Are there other relevant predictors?” So, what are other predictors? Please state in the last part of the abstract.

Response: Thanks, some of the relevant predictors for first stage modelling and all of the relevant predictors for second stage modelling are mentioned in the Results and Conclusion.

3. What is MSOA? All abbreviation should be defined first.

Response: Thanks, now defined at first mention in the Abstract.

4. The “discussion” part fits better with “conclusions”

Response: Thanks, now renamed as Conclusion.

Reviewer #2: INTRODUCTION

5. “Vaccination uptake is highly dependent on age, gender, ethnicity, and

deprivation, as is widely recognised 3 4 5 6 7 8 9” Are all the studies cited

from UK? Or, are they from different countries? Readers may assume they are all from

UK, I suggest author to breakdown each of the cited literatures.

Response: Thanks, I have separated these into UK and international studies.

6. “…in contrast with 41.1% in Liverpool and 67.4% in

Gloucestershire.” Where does this information come from? Please cite the

literature.

Response: Thanks, I have now cited the UK Government Coronavirus Dashboard as the source for the data from which these figures are derived.

7. Please define abbreviations: MSOA, UTLA, etc.

Response: Thanks, both are spelled out at first use in this section, with more detail to explain their meaning.

8. In the last paragraph, author is encouraged to elaborate as to why this approach

is appropriate. Is it more robust in comparison with other approaches?

Response: Thanks, I am not including this in the Introduction, but the Statistical methods section now refers to reviews of mixed models, and points out that it is overly restrictive to assume linearity. A comparison of GAM with other non-linear modelling techniques is beyond the scope of this paper, but as mentioned in the Discussion, “The Generalized Additive Model, implemented in R with the mgcv package, is an established technique for non-linear modelling.” For model D applied to the Third Injection, the Results show that the mixed model is superior (by anova, or the ML criterion or Proportion of Deviance Explained) to models using only the fixed effects, or only the random effect. 

Reviewer #2: METHODS

9. “Data sources” and “Statistical method and models” should

be under one section “Methods”

Response: Thanks, the section is now called “Materials and methods”, and contains subsections “Data sources” and “Statistical methods”. I think it is helpful to refer to these subsections separately.

Reviewer #2: RESULTS

10. Table 1 has no border, and should be revised. Not sure what this value mean -15317.25 (0.953), I understand that author put it in the caption, but that is not enough. Furthermore, why author does not make it uniform, e.g. Dose 1, Dose 2, Dose 3 or 1st injection, 2nd injection, 3rd injection?

Response: Tables are now formatted with borders, and include a Title and a Caption. Table 1 displays the ML value and the Proportion of Deviance Explained for each of the fitted models. These outputs are now explained in the supporting information S1_Appendix. The “dose” and “Third Injection” terminology follows that of the downloads from the UK Government Coronavirus Dashboard, and is explained in the “Data sources” subsection. Dose 1 and Dose 2 are the first and second primary vaccinations, but the Third Injection data comprises persons given either the Booster, or a 3rd primary given to clinically vulnerable persons.

11. Before Table 1, the paragraph only consists of 1 sentence. I encourage the author to move more sentences into the paragraph. And this applies for every data presented (including the following charts or tables).

Response: 11. Thanks, I have rewritten with longer paragraphs, in general.

12. “….except for distance to pharmacy (p<0.01).” Where can I see the data? Supplementary file? Author should cite it in the text.

Response: This sentence describes an aspect of the fitted output of model D applied to the Third Injection, using all the covariates. A summary of the output is now shown in S7_File.pdf

Reviewer #2: DISCUSSION

13. “…COVID-19 vaccine hesitant followed by the Pakistani/Bangladeshi group.” Provide citation please.

Response: This is a quote from the SAGE report in December 2020, as cited. I have moved the citation to just after the quote.

14. In the second paragraph of Discussion, consider to include these following studies.

# In a study performing willingness-to-pay analysis, flu vaccination is also among the positive predictors for COVID-19 vaccine acceptance. Citing: Sallam et al. Narra J 2022; 2(1): e74 – doi: 10.52225/narra.v2i1.74

# Side effects of COVID-19 vaccine become the major barrier for vaccination program in population from different countries. Citing: Rosiello et al. Narra J 2021; 1(3): e55- doi: 10.52225/narra.v1i3.55.

#Disinformation and certain religious narrative could discourage individuals to take COVID-19 vaccine. Citing: Hassan et al. Narra J 2021; 1(3): e 57 - doi: 10.52225/narra.v1i3.57

Response: Thanks, the Discussion now mentions two of the suggested studies.

15. I don’t understand what is the author trying to convey in the fourth paragph of the Discussion.

Response: This is now paragraph 6 of the Discussion, and concerns the fact that occupation also influences vaccine uptake. It cites data from the Office for National Statistics on how the proportion of the population receiving 3 vaccinations varied according to their employment, and also, separately, according to their ethnicity. For example, 80.4% of people employed in the Health Sector had received 3 vaccinations, as of 31 December 2021. The citations now go to the relevant earlier releases from the ONS.

16. This paragraph, “An ONS technical article…” Does author try to compare the method with previous study? I think this paragraph better fit in the introduction, rather than in discussion. Also, clarify the intention of putting this paragraph, because it seems author just arbitrarily put random paragraph in the Discussion.

Response: Thanks, I’ve amended this and placed it as the first paragraph in the subsection “Geographical factors”. It is included because although it was based on individual survey data, it also showed a significant regional effect.

17. Discussion needs a lot of re-arrangements to make it more readable and acceptable for publication. Other than re-arranging the paragraphs to optimize the logical flow, author is encourage to distinct the findings from previous studies and that of from present study, and followed by a clear comparison. At the present version, it’s very hard to get the comparative pictures of the studies.

Response: Thanks, I’ve re-ordered the presentation of previous research so that the discussion moves from surveys of individuals, to studies which show the impact of vaccination status on mortality and make recommendations about improving uptake, to studies which include geographical factors, including studies of childhood immunisation, one of which features the impact of change in public health budgets. The discussion then explains that my article concerns area data rather than individual data, and then comments on the results, some of which are compared with the previous research cited.

18. Please provide limitation of the study at the end of discussion.

Response: Thanks, I’ve included a paragraph on this at the end of Discussion.

19. Please provide a section of “Conclusions”

Response: Thanks, the final 3 paragraphs are now headed Conclusions.

---

## [Decision Letter · Decision Letter 1]

14 Jun 2022

Variation in COVID-19 booster uptake in England: an ecological study

PONE-D-22-05078R1

Dear Dr. Dropkin,

We’re pleased to inform you that your manuscript has been judged scientifically suitable for publication and will be formally accepted for publication once it meets all outstanding technical requirements.

Kind regards,

Harapan Harapan, MD, PhD

Academic Editor

PLOS ONE

Additional Editor Comments (optional):

Reviewers' comments:

Reviewer's Responses to Questions

**Comments to the Author**

1. If the authors have adequately addressed your comments raised in a previous round of review and you feel that this manuscript is now acceptable for publication, you may indicate that here to bypass the “Comments to the Author” section, enter your conflict of interest statement in the “Confidential to Editor” section, and submit your "Accept" recommendation.

Reviewer #1: All comments have been addressed

Reviewer #2: All comments have been addressed

2. Is the manuscript technically sound, and do the data support the conclusions?

Reviewer #1: Yes

Reviewer #2: Yes

3. Has the statistical analysis been performed appropriately and rigorously? 

Reviewer #1: Yes

Reviewer #2: Yes

4. Have the authors made all data underlying the findings in their manuscript fully available?

Reviewer #1: Yes

Reviewer #2: Yes

5. Is the manuscript presented in an intelligible fashion and written in standard English?

Reviewer #1: Yes

Reviewer #2: No

6. Review Comments to the Author

Reviewer #1: The author adequately responded to my previous comments.

Reviewer #2: The author has responded to all my concerns. The revised version is satisfactory and can be accepted for publication.

7. PLOS authors have the option to publish the peer review history of their article (what does this mean?). If published, this will include your full peer review and any attached files.

Reviewer #1: No

Reviewer #2: No

---

## [Editor Report · Acceptance letter]

20 Jun 2022

PONE-D-22-05078R1 

Variation in COVID-19 booster uptake in England: an ecological study 

Dear Dr. Dropkin:

I'm pleased to inform you that your manuscript has been deemed suitable for publication in PLOS ONE. Congratulations! Your manuscript is now with our production department. 

Kind regards, 

on behalf of

Dr. Harapan Harapan 

Academic Editor

PLOS ONE